# Voltage controlled Néel vector rotation in zero magnetic field

Ather Mahmood [1], Will Echtenkamp[1], Mike Street[1], Jun-Lei Wang[1], Shi Cao [1], Takashi Komesu[1], Peter A. Dowben [1], Pratyush Buragohain [1], Haidong Lu [1], Alexei Gruverman [1], Arun Parthasarathy [2], Shaloo Rakheja[3] & Christian Binek [1]✉

Multi-functional thin films of boron (B) doped $Cr_2O_3$ exhibit voltage-controlled and non-volatile Néel vector reorientation in the absence of an applied magnetic field, $H$. Toggling of antiferromagnetic states is demonstrated in prototype device structures at CMOS compatible temperatures between 300 and 400 K. The boundary magnetization associated with the Néel vector orientation serves as state variable which is read via magnetoresistive detection in a Pt Hall bar adjacent to the $B:Cr_2O_3$ film. Switching of the Hall voltage between zero and non-zero values implies Néel vector rotation by 90 degrees. Combined magnetometry, spin resolved inverse photoemission, electric transport and scanning probe microscopy measurements reveal B-dependent $T_N$ and resistivity enhancement, spin-canting, anisotropy reduction, dynamic polarization hysteresis and gate voltage dependent orientation of boundary magnetization. The combined effect enables $H = 0$, voltage controlled, nonvolatile Néel vector rotation at high-temperature. Theoretical modeling estimates switching speeds of about 100 ps making $B:Cr_2O_3$ a promising multifunctional single-phase material for energy efficient nonvolatile CMOS compatible memory applications.

[1] Department of Physics & Astronomy and the Nebraska Center for Materials and Nanoscience, University of Nebraska-Lincoln, Lincoln, NE, USA. [2] Department of Electrical Engineering, New York University, Brooklyn, NY, USA. [3] Holonyak Micro and Nanotechnology Laboratory, University of Illinois at Urbana–Champaign, Urbana, IL, USA. ✉email: cbinek@unl.edu

Magnetoelectric (ME) antiferromagnets, notably the archetypical ME insulator $Cr_2O_3$ (chromia), have long been exploited to realize voltage-controlled spintronic devices[1–3]. In contrast to their multiferroic counterparts[4,5], ME antiferromagnets have one spontaneous ferroic order parameter whose temperature ($T$) dependence determines the $T$-dependence of the ME response[6]. In pristine chromia, the linear ME response, $\alpha_{ij}$, sets in below the Néel temperature $T_N = 307$ K, where magnetization, $\underline{M}$, (polarization, $\underline{P}$) is induced by an electric $E$-field (magnetic $\overline{H}$-field) according to $\alpha_{ij} = \mu_0 \partial M_i / \partial E_j = \partial P_i / \partial H_j$[7]. Voltage-controlled switching of the Néel vector between 180° domain states can be achieved when their degeneracy is lifted. In chromia this is accomplished by simultaneously applying $E$ and $H$ along the easy axis ($c$-axis) such that the free energy difference $\Delta F = 2\alpha_{33} E_3 H_3$ overcomes the anisotropy energy barrier which separates the two antiferromagnetic (AFM) single domain states[8]. This mechanism has been exploited in voltage-controlled exchange bias (EB) heterostructures fabricated from chromia and an exchange coupled perpendicular anisotropic magnetic thin film such as CoPd[9–12]. Uncompensated AFM surfaces with equilibrium roughness often have a small interface magnetization. Therefore EB tends to be small in heterostructures based on antiferromagnets with uncompensated surfaces[13]. However, in ME antiferromagnets, an equilibrium magnetic moment associated with the AFM order parameter is symmetry allowed[14–16]. Note that due to the rigorous symmetry argument leading to boundary magnetization in linear ME antiferromagnets, the boundary magnetization is strictly tied to the orientation of the Néel vector. This moment can be sizable even in the presence of roughness and enables effective coupling with the magnetization of an adjacent ferromagnet. In such heterostructures, voltage-controlled reversal of the Néel vector switches the EB field between negative and positive values and can give rise to reversal of the remnant magnetization of the adjacent ferromagnet[9–12,17]. Note that switching based on the ME effect requires the simultaneous presence of $E$ and $H$ fields which, combined, have to overcome a critical product $(EH)_c$. The need for $H$-fields, even when provided via stray-fields or exchange fields[18], is detrimental to the implementation of practical devices. Despite this limitation, voltage-controlled EB heterolayers can serve as building blocks for voltage-controlled spintronics where the orientation of the exchange coupled magnetization defines the state variable[2]. Voltage-control promises energy efficient performance due to the absence of dissipative currents often required in spintronic devices which switch the state variable via Oersted fields, spin polarized electric currents through spin transfer torque[19], spin–orbit or Néel torque in metallic antiferromagnets[20].

Chromia is an insulating antiferromagnet whose ME properties can be exploited for voltage-controlled AFM spintronic devices. However, in its pristine form, chromia falls short of two prime objectives. Those are voltage-controlled switching in zero $H$-field and stable device operation above room temperature. High-temperature operation is an essential prerequisite to embed a ME device in a CMOS environment where typical load temperatures of central processing units reach 350 K or more[21].

Our work demonstrates that B-doping of chromia creates a single-phase material, which enables voltage-controlled nonvolatile rotation of the Néel vector in zero $H$-field and CMOS compatible operation temperatures. B:$Cr_2O_3$ simultaneously acquires new tunable functionalities in addition to the linear magnetoelectricity known from $Cr_2O_3$. Those include $T_N$ and resistivity enhancement, reduced and voltage controllable anisotropy, spin-canting and transient electric polarizability not observed in pure chromia. Our data imply that B-doping breaks local symmetry allowing for the formation of polar nanoregions (PNRs). The PNRs orient in an applied electric field giving rise to

transient polarization and piezoelectricity. Piezoelectrically induced strain alters the reduced, nearly cubic anisotropy[22] of B:$Cr_2O_3$ giving rise to a $\pi/2$ rotation of the Néel vector into a new stable state. Coupling between the AFM order parameter and the boundary magnetization allows to read out the AFM state via magnetoresistive detection in an adjacent Pt Hall bar. This structure demonstrates the fundamental building block of an ultra-low power electrically controlled AFM spintronic device which operates above room temperature and in zero applied magnetic field.

## Results

**Multi-functional properties of B:$Cr_2O_3$.** In ME antiferromagnets, the AFM order and ME response disappear at $T > T_N$ rendering device operation near or above $T_N$ impossible. Chromia's Néel temperature of 307 K is insufficient for CMOS compatible applications. To mitigate this shortcoming, B-doping has been established as a viable path to increase $T_N$[23,24] with its only potential alternative being epitaxial or chemical straining[22,25,26]. Although AFM order and the persistence of magnetoelectricity have been predicted and demonstrated up to $T = 400$ K in B-doped chromia[23,27], utilizing the high-$T_N$ material in device structures which rely on EB, remains elusive. In device architectures based on voltage-controlled EB, B-doping does not translate into improved device performance. Although B-doping can increase the blocking temperature, exchange coupling is not accompanied by effective pinning and thus voltage-control of the ferromagnet (see note 1 in the Supplementary information). We attribute the detrimental effect of B-doping on perpendicular EB to reduced magnetic anisotropy and canting of the interface magnetization relative to the surface normal. Nevertheless, AFM order and ME functionality are preserved above 307 K, as is evident from magnetometry[23], spin polarized inverse photoemission spectroscopy, and X-ray magnetic circular dichroism (see Fig. S11 in the Supplementary information), giving rise to alternative device concepts with reduced complexity[3].

When growing B:$Cr_2O_3$/CoPd heterolayers with in-plane anisotropic CoPd films, sizable EB reappears below the blocking temperature substantiating the interpretation that B-doping is associated with anisotropy reduction and spin canting. Figure 1 shows the complex $T$-dependence, $\mu_0 H_{EB}$ versus $T$, of the EB field in $Cr_2O_{2.9}B_{0.1}$(100 nm)/Pd(0.5 nm)Co(3 nm)Pd(0.5 nm), where the ratio of Co to Pd film thickness tunes the anisotropy of CoPd heterolayers to be in the plane (see note 9.3 in the Supplementary information). The EB field is non-zero up to 400 K but only sizable at $T < 100$ K (open squares in Fig. 1). Insets a and b in Fig. 1 show representative in-plane CoPd hysteresis loops measured via vibrating sample magnetometry. The loops show positive EB fields of 0.18 and 1.1 mT at 323 and 232 K, respectively. The absence of perpendicular EB and the presence of in-plane EB indicate that the boundary magnetization in B-doped chromia tends to tilt away from the $c$-axis in agreement with recent findings in $CoFe_2O_4$–$Cr_2O_3$ nanocomposites[28]. There is strong support for this interpretation from $T$-dependent spin-resolved inverse photoemission shown in Figs. 1c and S11c. This result was seen to be consistent with X-ray magnetic dichroism which also found an in-plane component of $Cr^{3+}$ magnetic moment (see Fig. S11a and b in the Supplementary information). Inverse photoemission is a surface sensitive probe of the unoccupied spin-dependent states. The presence of a spin-resolved inverse photoemission signal, in the geometry of our experiment requires a tilt of the surface spins relative to the $c$-axis, as indicated in the cartoon in Fig. 1c. The difference in intensity for each spin polarization versus binding energy (blue up and red down triangles in Fig. 1c are signals associated with spin up and

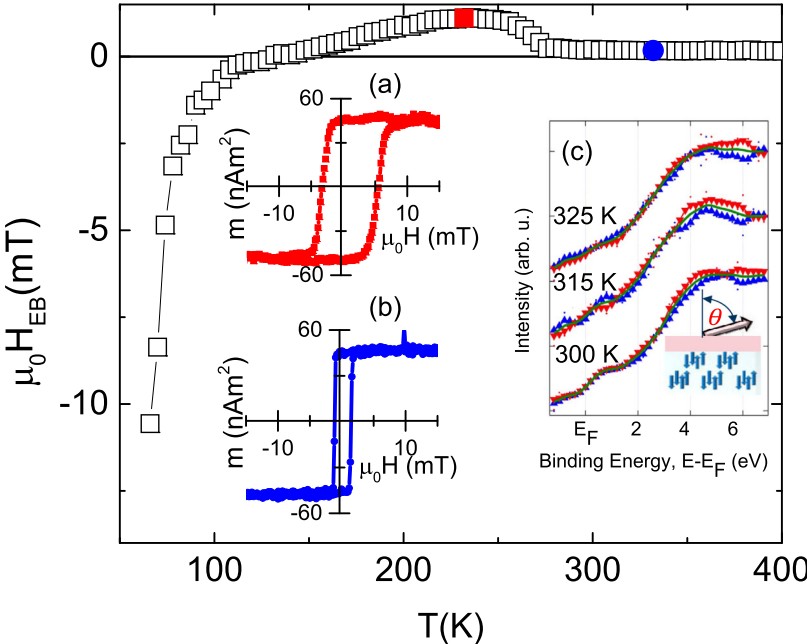

**Fig. 1 Spectroscopy and magnetometry of B-doped chromia films and their heterostructures.** EB field $\mu_0 H_{EB}$ versus $T$ (open squares in main panel) for heterolayer based on B-doped chromia and in-plane anisotropic CoPd top layer. Insets **a** (red squares) and **b** (blue circles) show representative hysteresis loops at 323 and 232 K. The loops are associated with $\mu_0 H_{EB}$ versus $T$ data highlighted by solid symbols. Inset **c** shows spin-resolved inverse photoemission data for the surface of a B-doped chromia thin film at 300, 315, and 325 K. The inset shows a cartoon of the film with spin structure (tilt not shown) and boundary magnetization tilted by the angle $\theta$ relative to the $c$-axis (surface normal).

spin down electronic states) at 300, 315, and 325 K indicates that boundary magnetization and thus AFM order persist above the Néel temperature of pure chromia. The reduction of magnetic anisotropy and the associated canting of the boundary magnetization in $B:Cr_2O_3$ hamper its use in devices which rely on exchange coupling between a FM layer and the ME antiferromagnet. In Hall bar structures, where the FM constituent layer has been eliminated, reduced anisotropy becomes a beneficial feature. Hall bar structures from nonmagnetic heavy metals on top of chromia can sense boundary magnetization and serve as readout components in all AFM memory devices[29]. In our Hall-device, a Pt Hall bar detects a transverse voltage signal, $V_{xy}$, in response to an in-plane current density **j** (see Fig. 2a for Hall geometry)[30–32]. The Hall-like signal $V_{xy}$ is widely believed to originate from spin Hall magnetoresistance due to the fact that the mixing conductance has a non-zero imaginary component[31,33]. Potential additional contributions might originate from the anomalous Hall effect caused by magnetization which is proximity induced in the heavy metal Hall bar by the exchange field of the boundary magnetization or by anomalous Hall effect generated by spin chirality[32,34]. Regardless of the details of the mechanisms generating $V_{xy}$, it has been experimentally established that $V_{xy}$ is a reliable proxy for the orientation of AFM boundary magnetization[30,31,35]. Of particular importance for this interpretation in heavy metal/chromia bilayers are the complementary studies of magnetotransport and direct imaging of boundary magnetization. In chromia, x-ray magnetic circular dichroism photoemission electron microscopy (PEEM) is the method of choice to detect Néel vector reversal via reversal of the AFM boundary magnetization[36]. In antiferromagnets such as $CuMnAs$ and $Mn_2Au$ direct measurement of the reorientation of the Néel vector require x-ray magnetic linear dichrosim-PEEM[37,38].

An optical image of a Hall bar device which detects magnetization states of the electrically controllable boundary magnetization is shown in Fig. 2b. The device comprises a

5 nm-thick Pt Hall-cross with two orthogonal legs of 7.0 μm × 1.0 μm. The Hall cross was fabricated by depositing Pt via DC magnetron sputtering, and subsequent lithographic patterning, on top of a 200 nm B-doped chromia film. The B-doped chromia has been grown via pulsed laser deposition (PLD) on top of 20 nm PLD-grown $V_2O_3$ (see note 9.1 in the Supplementary information). The $V_2O_3$ film serves as bottom electrode due to its metallicity above the insulator to metal transition at $T \approx 150$ K[39]. Reduced device complexity through absence of an exchange coupled FM layer ensures that reorientation of the Néel vector is not affected by interfacial exchange energy which increasingly contributes to the switching energetics in EB-type heterostructures on scaling down the device thickness[12,35,40].

Figure 2c and d shows $V_{xy}$ of the Hall measurements taken at $T = 300$ K with a readout current of 20 μA. The sequence of measurements is numbered and the Hall signal $V_{xy}$ is shown as a function of this numbering. Hundred subsequent Hall measurements in zero field were performed to determine a baseline for $V_{xy}$ associated with a particular AFM state. After every hundredth point, a voltage pulse of $V_G = \pm 25$ V was applied across the AFM film for a duration of $\Delta t = 4$ s. The applied voltage gives rise to an electric field, $E$. After exposing the AFM film to an $E$-field, 100 subsequent data points are taken at $E = 0$. Each of the 100 measurements takes about 400 ms probing the nonvolatile state of the antiferromagnet. The grid of dashed vertical lines marks the points where unipolar voltage pulses are applied. The data in Fig. 2c demonstrate that a voltage pulse, $V_G$, can switch $V_{xy}$ and thus the AFM spin structure of the B-doped chromia film in zero applied magnetic field. The control experiment shown in Fig. 2d shows data taken at $\mu_0 H = -1$ T implying that the signal switching is independent of the presence of an applied magnetic field. Figure 2a shows a scheme of the Hall-bar device associated with the optical image of Fig. 2b. The Pt Hall-bar and the bottom $V_2O_3$ film serve as electrodes allowing to apply the gate voltage $V_G$ across the AFM film. The spin structure of chromia is depicted by up (green) and down (blue) spins. For simplicity,

canting present in B-doped chromia as well as the fact that the magnetic unit cell contains four sublattices are not shown.

Remarkably, the toggling between AFM domain states in $H = 0$ and $\mu_0 H = -1$ T indicates that ME switching, which is the well-established switching mechanism in pure chromia, can be ruled

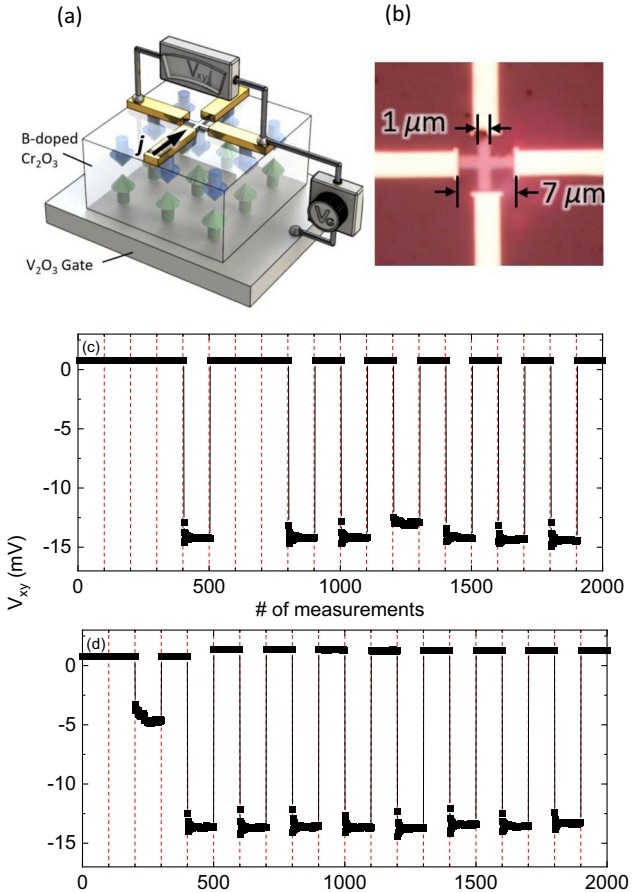

**Fig. 2 Image of the lithographically patterned device, illustration of the experiment, and switching results. a** Cartoon of Hall bar device showing $V_2O_3$ back gate, B-doped $Cr_2O_3$ film with AFM spin structure, Pt Hall cross with Au electrodes, current density $j$ flowing in direction of black arrow causing signal $V_{xy}$ which depends on applied voltage $V_G$. **b** Optical image of the device with 7.0 μm × 1.0 μm legs forming the Pt Hall cross with attached Au electrodes on top of the B-doped $Cr_2O_3$ film. **c** and **d** show $V_{xy}$ versus the number (#) of measurements. Vertical dashed lines indicate respective application of a voltage pulse $V_G = \pm 25$ V. Measurements are done at $T = 300$ K in (**c**) $H = 0$ T, and **d** $\mu_0 H = -1$ T applied magnetic field.

out. In both cases ($H = 0$ and $\mu_0 H = -1$ T), deterministic and nonvolatile switching between two distinct AFM states is observed. The high degree of asymmetry in $V_{xy}$ on switching between different nonvolatile AFM states associated with $V_{xy} \approx 0$ and $V_{xy} \approx -15$ mV implies 90° rotation of the Néel vector in sharp contrast to Hall-signals observed for 180° switching[29]. Rotation of the Néel vector by $\pi/2$ is consistent with the fact that time reversal symmetry is not broken by an electric field. Equilibrium reversal of the Néel vector by $\pi$ requires a field combination such as $E \cdot H$ which breaks time inversion symmetry.

The findings presented in Fig. 2 are supported by hysteresis loops $V_{xy}$ versus $V_G$ measured at $T = 300$ K in $H = 0$ (Fig. 3a open squares) and $\mu_0 H = -1$ T applied along the $c$-axis of the sample (Fig. 3b solid squares). The voltage $V_G$ is applied as a quasistatic pulse between top and bottom electrode of the device (see sketch in Fig. 2a) and removed prior to probing the transverse voltage $V_{xy}$. The sharp transitions at the coercive voltages of about ±15 V resemble deterministic switching between distinct AFM states. The magnetic field has virtually no effect on the voltage-controlled transition. We confirmed these results for several devices and various temperatures (see note 3 and Fig. S4 in the Supplementary information for loops at $T = 330$ and 400 K). Because the switching is triggered by the applied electric field alone we exclude ME switching and investigate an indirect coupling between induced polarization and AFM order[41]. Although the magnetic field has no effect on switching, it does affect the Hall voltage. The dissimilar field dependence of $V_{xy}$ in the two AFM states provides strong support for the magnetic origin of the switching shown in Figs. 2 and 3 and allows the assignment of the orientation of the boundary magnetization to the distinct voltage states (for details see Fig. S5 in the Supplementary Note 3).

**Independent evidence for electrically controlled magnetism in B:$Cr_2O_3$.** To provide independent evidence that the switching effects shown in Figs. 2 and 3 are magnetic in origin, we carried out additional measurements by magnetic force microscopy (MFM). MFM utilizes the long-range forces, originating from interaction between a magnetized tip and the magnetic stray field of the sample. Figure 4 shows the topographic and MFM images of a segment of the Pt Hall cross, which was deposited on the same B:$Cr_2O_3$ thin film used for the devices in Figs. 2 and 3. The MFM images illustrate a change in the magnetic response signal of the pristine sample after application of the poling pulses of 2 s duration and ±10 V magnitude. Clearly visible is the drastic reduction of the MFM contrast after application of the +10 V pulse and its partial recovery after application of the −10 V pulse. Kelvin probe force microscopy measurements (Fig. S6 in the

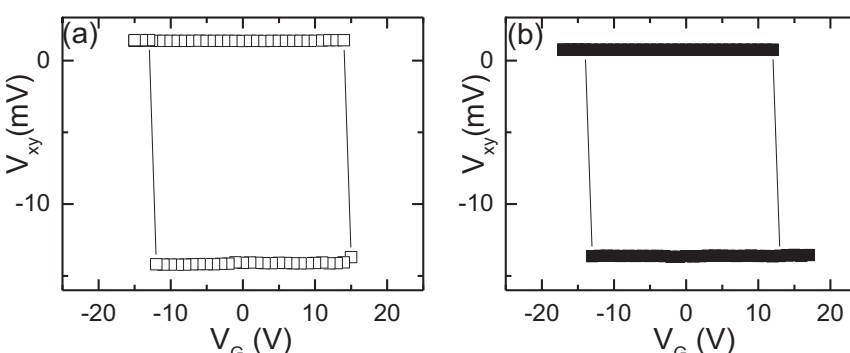

**Fig. 3 Electric hysteresis of Hall-like signal.** $V_{xy}$ versus $V_G$ hysteresis loops measured at $T = 300$ K in **a** $\mu_0 H = -1$ T (open squares) and **b** $H = 0$ (solid squares).

Supplementary information) allowed us to rule out the effect of the parasitic electrostatic tip–sample interaction signal on the observed MFM contrast variations.

Going further, we are able to provide experimental evidence for the presence of transient dielectric polarization in B: $Cr_2O_3$ and show in a control experiment (see Fig. S8 in the Supplementary Information) that no such response is observed in pure chromia. This rules out the possibility of an artifact in the PFM measurements. In order to explain the qualitative dielectric differences between pure and B-doped chromia it is worth to mention that our B-doped samples show significantly enhanced resistivity compared to films of pure grown $Cr_2O_3$ grown in the same PLD chamber. The resistivity enhancement directly manifests in electric transport measurements (Fig. S9 in the Supplementary information) but is even more compelling when comparing the inverse photoemission data of pure and B-doped chromia (see Fig. S2 in the Supplementary Information) where smearing of the intensity profile near the Fermi energy is indicative of charging and thus increased resistivity. Some studies reported similarly high resistivity in thin chromia films doped by metallic ions such as $Ti^{4+}$ [42]. Enhanced resistivity in B: $Cr_2O_3$ is a necessary prerequisite for the presence of polar states. Their formation require strong internal electric fields at low leakage current.

**Probing-induced polarization in B:$Cr_2O_3$.** By using piezo-response force microscopy (PFM), we provide experimental evidence for the induced dielectric polarization in B:$Cr_2O_3$. In PFM, the electromechanical response measured locally in the region underneath the tip can be associated with the polar state of the material[43]. PFM spectroscopic measurements of the Pt/B:$Cr_2O_3$/ $V_2O_3$ heterostructure performed in the bias-off regime reveal typical butterfly-shape amplitude hysteresis loops along with the 180° change in the PFM phase (Fig. 5a) indicating presence of the switchable polarization in B:$Cr_2O_3$. Importantly, no such response has been observed in pure chromia (Fig. S8 in the Supplementary information). Given that these measurements have been carried out on the Pt top electrode, the electrostatic contribution to the measured PFM signal can be ruled out.

Previously, it has been shown that robust PFM signals could be registered in the non-ferroelectric materials due to the electrically induced polarization, which in turn could result from various mechanisms, such as redistribution of oxygen vacancies[44] or reorientation of PNRs[45]. Figure 5b shows that the PFM amplitude signal, measured in the Pt/B:$Cr_2O_3$/$V_2O_3$ heterostructure, is not stable but relaxes logarithmically with a characteristic relaxation time varying in the range from several hundreds of milliseconds to tens of seconds. This behavior suggests that voltage pulse application produces a metastable polarization state in B:$Cr_2O_3$. Similarity of the PFM relaxation dynamics to that observed in ferroelectric relaxors allows us to assume that the induced polarization in B:$Cr_2O_3$ could be due to the alignment of the naturally existing PNRs (see discussion below).

The effective $d_{33,eff}$ piezoelectric coefficient of B:$Cr_2O_3$ can be estimated by comparing its electromechanical response with the PFM signal detected in another material with the well-known piezoelectric properties[46,47] (see note 7 in the Supplementary information for details). In our case, we chose lithium niobate $LiNbO_3$ as a reference material. Comparative analysis of the PFM amplitude signals measured in B:$Cr_2O_3$ and $LiNbO_3$ yields a value of ~8 pm/V for the effective $d_{33}$ coefficient in B:$Cr_2O_3$ right after +7 V pulse application.

The induced polarization is a key ingredient for the voltage-controlled Néel vector rotation. Its origin and indirect coupling with the AFM order parameter is discussed next.

## Discussion

Our experiments show multifaceted effects of B-doping on magnetic and dielectric properties of $Cr_2O_3$ which, when acting jointly, enable the $H = 0$, nonvolatile Néel vector rotation above room temperature. The logarithmic relaxation of the induced piezo-response after poling is characteristic of thermally activated

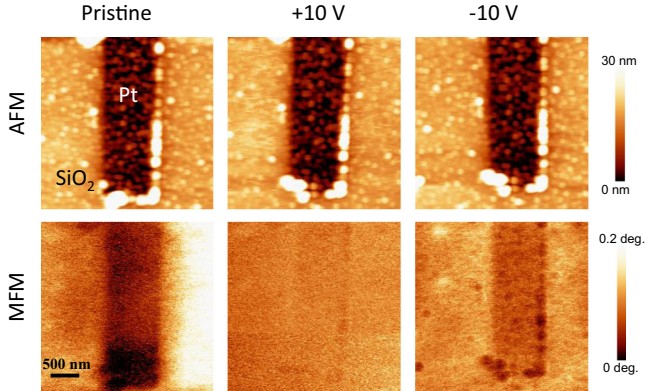

**Fig. 4 Topographic and magnetic surface characterization in response to applied voltages.** Top row: topographic images of a segment of the Pt Hall cross before and after application of the poling pulses (2 s; ±10 V). Bottom row: MFM images of the same segment after application of the poling pulses.

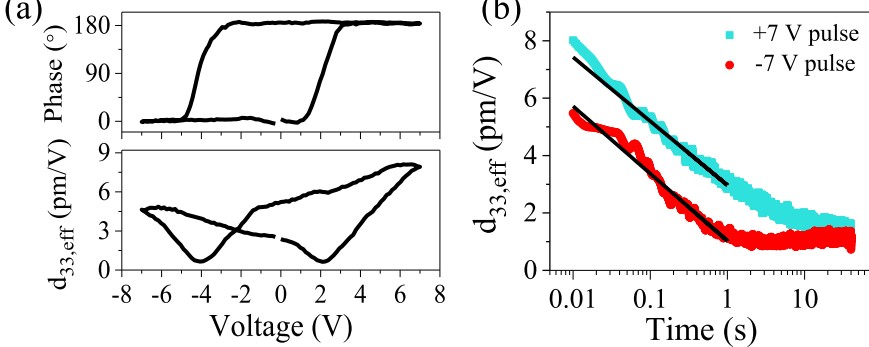

**Fig. 5 Voltage and time dependence of local piezo response. a** Bias-off PFM amplitude (bottom panel) and phase (top panel) hysteresis loops measured in the Pt/B:$Cr_2O_3$(200 nm)/$V_2O_3$ structure. **b** PFM amplitude signal as a function of time elapsed after application of a positive (blue) and negative (red) poling pulses (7 V, 12.5 ms). Solid lines in **b** illustrate the logarithmic fit of the temporal decay of the effective piezoelectric coefficient $d_{33}(t)$ in accordance with a relaxation model for electric field-oriented PNRs.

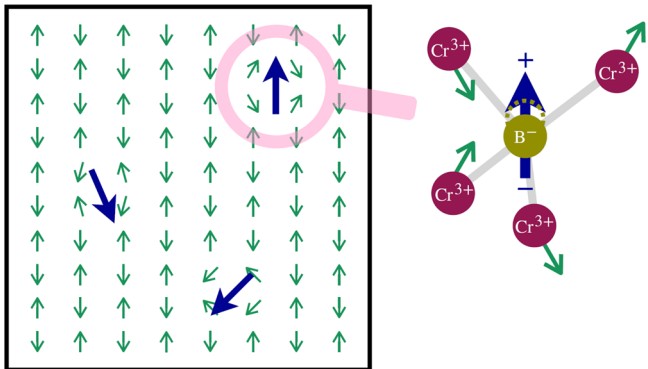

**Fig. 6 Illustration of polar nanoregions.** Local polarization (blue arrow) arises from off-center B⁻ substitutions and accompanying inhomogeneous local strains which give rise to canting of neighboring spins.

PNRs[48] and contrasts stationary polar order in multiferroics where spontaneous polar and magnetic order coexist[49]. Figure 6 illustrates a possible mechanism which gives rise to PNRs in B: $Cr_2O_3$. The PNRs are linked to inhomogeneous local strains produced by random substitution of O atoms for B. The local strain moves the B atom to an off-center position within $BCr_4$ tetrahedra resulting in emergence of PNRs[50,51]. Electric field-induced alignment of dipole moments in PNRs leads to detectable piezoresponse. Thermal fluctuations of polarization after the field is off cause temporal decay of the piezoelectric coefficient, which follows the logarithmic law: $d_{zz}(t) \approx d_0 - S \log_{10}\left(\frac{t}{t_0}\right)$ (Fig. 5b) where the slope $S = 2.2 - 2.3$ is a measure of viscosity and $\frac{1}{t_0} = 1$ Hz is the attempt frequency. The observed difference in the peak amplitude of the piezo-response between positive and negative poling voltages may be caused by residual strain near the surface[48]. Similar logarithmic time dependences such as those seen in the piezo-response in Fig. 5b have been reported in the case of magnetic after effects, where magnetic viscosity originates from the distribution of activation energy barriers of ferromagnetic domains[52].

*E*-field-induced orientation of PNRs plays a critical role in the interplay between polarization, piezoelectricity, strain and anisotropy control, which gives rise to the Néel vector rotation. The *E*-field aligns the PNRs. This process transforms the local strain distribution into a uniform strain field $\epsilon = d_{zz}E$. The strain alters the perpendicular magnetic anisotropy $K_\perp$ via magnetoelastic coupling. Before we investigated the effect of the magnetoelastic coupling on B:$Cr_2O_3$, we reconsidered the case of undoped $Cr_2O_3$. Chromia has perpendicular uniaxial anisotropy and collinear alignment of the spins along the *c*-axis. In a (0001)-oriented $Cr_2O_3$ film the *c*-axis is normal to the film plane. However, stable in-plane domain variants can be produced through lattice mismatch with a substrate[53] or substitutional defects. The chromia film surface itself can be considered as a defect where temperature-driven surface reconstructions take place, which include spin-canting effects[54–56]. First-principles calculations show that the magnetic anisotropy in unstrained $Cr_2O_3$ is close to cubic and the preference to perpendicular anisotropy is strongly modified by strain as $\Delta K_\perp/K_\perp \approx 10^3 \epsilon$ [22]. Magnetometry and inverse photo-emission data (see Fig. 1) provide experimental evidence that the anisotropy in B:$Cr_2O_3$ is reduced and canting at the surface is more prominent compared to undoped chromia. We therefore express the free energy of the B-doped $Cr_2O_3$ in terms of the perpendicular and in-plane components, $n_\perp$ and $n_\parallel$, of the Néel vector as $F_{aniso} = K_0 n_\perp^2 n_\parallel^2 - K_\perp n_\perp^2$, where $K_0 \gg K_\perp$. Depending on the direction of electric *E*-field (parallel or antiparallel to the *c*-axis), the piezoelectrically induced strain is compressive or

tensile enabling rotation of the Néel vector between in-plane and perpendicular orientations in accordance with the switching condition $|\Delta K_\perp/K_\perp| = 1$ (see note 8 and Fig. S10 in the Supplementary information for details). The nearly cubic anisotropy creates local minima in the free energy landscape giving rise to nonvolatility after removal of the *E*-field. This switching criterion is in agreement with the experimentally observed coercive electric field of $E_c \approx 15$ V/200 nm for a piezo-response coefficient of $d_{zz} = 13$ pm/V. A $d_{zz}$ on the order of a few pm/V is in agreement with our estimate obtained from the PFM data (Fig. 5) and supports the model.

Although the magnetoresistive switching data shown in Figs. 2 and 3 refer only to quasistatic experimental conditions it is possible to estimate the switching speed for nonvolatile memory devices based on the outlined mechanism. The equation of motion for 90° reorientation of the Néel vector maps onto angular relaxation of a damped gravitational pendulum from an inverted position[57]. Using this model we estimate a speed of $10M_s/(\alpha\gamma J) = 100$ ps for 90° planar relaxation of the Néel order, where $M_s = 1.9 \times 10^5$ A/m is the sublattice magnetization, $\alpha = 2 \times 10^{-4}$ is the Gilbert damping, $J = 9.5 \times 10^7$ J/m³ is the intersublattice exchange and $\gamma$ is the electron gyromagnetic ratio. This underlines the advantage of AFM spintronics outperforming device concepts which rely on notoriously slow reversal of ferromagnetic constituents taking place on the ns time scale determined by the inverse Larmor precession frequency.

Energy-efficient isothermal and deterministic switching of remnant magnetic states above room temperature and in the absence of a magnetic field is fundamentally challenging and of highest interest for ultra-low power memory and logic device applications. Switching of AFM states is of particular interest due to inherent switching speed advantages over ferromagnets and robustness against external magnetic field perturbations. The work presented here demonstrates that the prototypical magnetoelectric antiferromagnet $Cr_2O_3$ can be tuned into a multi-functional high-$T_N$ material through B-doping. Emerging functionality associated with B-doping include purely electric-controlled 90° nonvolatile rotation of the Néel vector up to $T = 400$ K. Indirect coupling between polar and AFM order explains the experimental findings and allows to estimate switching speeds on the order of 100 ps. An energy efficient memory which operates up to 400 K in zero magnetic field has been fabricated on the basis of B-doped chromia. Future investigations will focus on device stability, switching speed, and scaling.

## Methods

Magneto-optical polar Kerr effect and vibrating sample magnetometry are utilized to investigate heterolayers of B-doped $Cr_2O_3$ (chromia) and ferromagnetic CoPd with tuned perpendicular and in-plane anisotropy for temperature-dependent EB. Magnetometry together with temperature-dependent spin-polarized inverse photoemission spectroscopy were utilized to investigate canting of the AFM interface magnetization relative to the film normal.

Our spin-polarized inverse photoemission experiments utilize a transversely polarized spin electron gun based upon the Ciccacci design[58]. As described elsewhere[23,58], the spin electron gun was used in combination with an iodine-based Geiger–Müller isochromat photon detector with a $SrF_2$ window. As is typical of such instruments, the electron gun has 28% spin polarization, and the data have been corrected for this incident gun polarization. The direction of electron polarization is in the plane of the sample. The magnetoelectric cooling was accomplished in an axial magnetic field of >40 mT and a voltage of 3 kV applied across the film thickness. The Fermi level was established from tantalum and gold foils in electrical contact with the sample.

The AFM constituent films of the EB heterostructures are grown by PLD while the anisotropy tuned CoPd multilayers are deposited via molecular beam epitaxy. PLD in ultra-high vacuum is used to grow (0001)-oriented films of the sesquioxides $V_2O_3$ and subsequently, in the presence of a decaborane background gas, B-doped chromia. The $V_2O_3$ film, which is grown on the *c*-plane of a sapphire single-crystalline substrate, serves as bottom electrode in a gated Hall bar structure. A 5 nm Pt film is deposited via DC magnetron sputtering on top of the B-doped chromia film and various Hall crosses are lithographically patterned. The Hall cross

serves as readout device which detects the voltage-controlled AFM states of the chromia film.

### Electrical measurements

*PFM characterization.* PFM spectroscopy loops were obtained in the resonance tracking mode using a commercial atomic force microscope system (MFP-3D, Asylum Research) by employing Pt-coated tips (HQ:NSC18/Pt, Mikromasch). The AC driving voltage was varied between 0.75 and 1 V at a frequency of ~330 kHz. For the PFM studies, capacitive devices were fabricated on the same samples as were discussed throughout this manuscript. Focused Ion Beam was used to mill $2 \times 2 \, \mu m^2$ sized capacitive structures. In all PFM measurements, the top electrode was biased while the bottom electrode grounded.

*Theory.* The switching criterion has been derived analytically by variation of the anisotropy energy. The switching speed of a thermally excited state has been determined by the time it takes for the state to relax from the top of the energy barrier to the bottom. The relaxation process has been mapped to damping of an equivalent inverted pendulum.

## Data availability

All data that support the plots within this paper and other findings of this study are available from the corresponding authors upon reasonable request.

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

## Acknowledgements

We gratefully acknowledge financial support by the Army Research Office through the MURI program under Grant Number W911NF-16-1-0472. This work was supported in part by nCORE, a wholly owned subsidiary of the Semiconductor Research Corporation (SRC), through the Center on Antiferromagnetic Magneto-electric Memory and Logic tasks #2760.00 and #2760.002 and NSF through ECCS 1740136, the Nebraska Materials Research Science and Engineering Center grant No. DMR-1420645, and the Nebraska Nanoscale Facility: National Nanotechnology Coordinated Infrastructure and the Nebraska Center for Materials and Nanoscience, under Award ECCS: 1542182, and the Nebraska Research Initiative. Magnetic characterization was in part performed at the NanoEngineering Research Core Facility (NERCF), which is partially funded by the Nebraska Research Initiative.

## Author contributions

A.M., W.E., and C.B. designed the study. A.M., W.E. fabricated the samples and conceived the electric transport measurements and magnetic characterization. A.M. lithographically patterned the Hall bar structures. M.S. did magnetometry measurements on exchange bias heterostructures based on B-doped chromia. J.-L.W. designed the 3-dimensional illustration of the Hall device and implemented the computer control of the transport experiments. S.C., T.K. performed the spin-resolved inverse photoemission experiments. P.A.D. directed and conceived the inverse photoemission and XMCD experiments. P.B. and H.L. performed the scanning probe microscopy (SPM) measurements. A.G. directed and conceived the SPM measurements. A.P. and S.R. developed the theoretical framework. C.B. directed the overall study. All authors contributed to the scientific process and the refinement of the manuscript.

## Competing interests

The authors declare no competing interests.
