## [Peer Review File · Nature Communications]

Reviewers' Comments:

Reviewer #1:

Remarks to the Author:

The manuscript describes a voltage-induced switching of Hall resistance in B-doped Cr₂O₃/Pt bilayers at room temperature. The work assumed that the Hall resistance changes are related to the switching of antiferromagnetic moments. Basically the motivation is interesting. But there are two main problems for the work, the first is the novelty taking relevant publications into account; the second is that the conclusion is well supported by the data. Thus I do not recommend its publication in *Nature Communications* with a high standard.

1. The first concerning is the justification of the relationship between Hall resistance and antiferromagnetic moments. Typically, antiferromagnetic materials show the spin-flop transition, where the antiferromagnetic moments could be efficiently rotated under a large magnetic field. Therefore, I suggest the author could give more evidence that the voltages induced-Hall resistance variations are distinct under different magnetic fields.
2. The authors claim that "the switching of the antiferromagnetic order parameter is reflected in the transverse Hall voltage", so they ascribe the variation of V_{xy} observed in their experiments to the switching of the antiferromagnetic order. However, the conclusion is not supported by the experiments. The variation of the V_{xy} can also be caused by other artifacts. Therefore, as suggested by the reviewer #1, some direct evidence may be imperative to support the conclusion.
3. The authors mentioned the CMOS-compatible temperatures in the title, i.e. 300K to 400K. However, most of the experiments are conducted at room temperature. Thus maybe the title can be changed from "...at CMOS-compatible temperatures" to "... at room temperature".
4. The author ascribe the " $V_{xy} \approx 0$ and $V_{xy} \approx -15\text{mV}$ " to 90-degree rotation of the Néel vector, why?
5. Spin Hall resistance in the B-doped Cr₂O₃/Pt system should be studied before the voltage controlled experiments.
6. To better understand the mechanism of the variation of V_{xy} , the authors should present more control experiences. For instance, when the oxide is not B-doped Cr₂O₃, but other AFM or even non-magnetic materials, whether the resistance variation still exists.
7. Besides, the voltage controlled Néel vector rotation has been realized in many previous works.

Reviewer #3:

Remarks to the Author:

Authors report the voltage-driven switching of the magnetoresistive resistance in Pt on B-doped Cr₂O₃. They attributed to the results to the piezoelectric switching of the polar nanoregion (PNR) and the accompanied rotation of boundary magnetization. I recognize the effort to revise the manuscript from the previous version. But still, there are some important concern on the proposed mechanism. Although various properties of the B-doped Cr₂O₃, some of them are heart of this paper, are known on the analogy of the un-doped Cr₂O₃, it is not obvious. Based on them, I think the paper is not acceptable in the present form.

As pointed out by reviewer 1, authors attribute the resistive switching to the rotation of the boundary magnetization. Authors claim that the coupling between the boundary magnetization and

the Hall effect should be robust for the B-doped Cr₂O₃, but it seems that this claiming is just the speculation from the analogy of the un-doped Cr₂O₃. Besides, the mechanism of the hall effect of Pt on Cr₂O₃ is under the debate. For example, other than the boundary magnetization, there are some reports proposing other mechanism such as Spin Hall effect (R. Schulitz et al., Appl. Phys. Lett. 112, 132401 (2018).), and interfacial spin chirality (T. Moriyama et al., Phys. Rev. Applied 13, 034052 (2020).) To ensure the proposed mechanism, the direct evidence to show the switching of the Néel vector and the boundary magnetization by the electric field.

In addition, as discussed in the un-doped Cr₂O₃, the boundary magnetization of the partial effect of the magnetoelectricity of Cr₂O₃. However, I cannot find the robust evidence that the B-doped Cr₂O₃ also show the similar magnetoelectricity to the un-doped Cr₂O₃. As pointed out in the manuscript, many of physical/structural properties such as magnetic anisotropy, spin orientation, local circumstance of Cr³⁺ are different from the un-doped Cr₂O₃. Then, the similar magnetoelectricity should not be ensured in the B-doped Cr₂O₃. For example, for the B-doping, the valence state of Cr ion can be change to keep the electrical neutrality. Thus, authors have to show that the B-doped Cr₂O₃ actually show the magnetoelectricity similar to the un-doped Cr₂O₃ and the existence of the boundary magnetization on the B-doped Cr₂O₃(0001).

Letter Legend: Referee comments are given in **Blue**, our responses in **Black** and the corresponding changes to the manuscript are in **Green**.

Figure legend: Figures, which appear in the reviewer reply only are labeled as “R”. Figures added to the main text are labeled by numbers only and figures added to the supplementary information are labeled by “S”.

Reviewers' comments:

Reviewer #1 (Remarks to the Author):

The manuscript describes a voltage-induced switching of Hall resistance in B-doped Cr₂O₃/Pt bilayers at room temperature. The work assumed that the Hall resistance changes are related to the switching of antiferromagnetic moments. Basically the motivation is interesting. But there are two main problems for the work, the first is the novelty taking relevant publications into account; the second is that the conclusion is well supported by the data. Thus I do not recommend its publication in Nature Communications with a high standard.

Reply:

We appreciate that the reviewer finds the motivation of our work interesting. Next we address the concerns of the reviewer point by point.

1. The first concerning is the justification of the relationship between Hall resistance and antiferromagnetic moments. Typically, antiferromagnetic materials show the spin-flop transition, where the antiferromagnetic moments could be efficiently rotated under a large magnetic field. Therefore, I suggest the author could give more evidence that the voltages induced-Hall resistance variations are distinct under different magnetic fields.

Reply: We agree with the reviewer that the suggested spin-flop experiment would be beneficial to add evidence that the Hall resistance signal is a proxy for the orientation of the Néel vector and thus the boundary magnetization. When extrapolating from pure Cr₂O₃, the spin-flop experiment is not practical in a laboratory transport setup because the spin-flop field exceeds 12 T at 300 K (Physica B: Condensed Matter **204** (1), 292-297 (1995)). However, we have designed an experiment which shows that voltage induced Hall resistance variations are clearly distinct under moderate magnetic field variation between -1 and +1 T. Before outlining the details of this experiment, we like to point out that there is also a bulk of literature showing that, in the case of Cr₂O₃/Pt Hall bar devices, the Hall resistance is associated with the orientation of the boundary magnetization. For example, in Applied Physics Letters **112** (13), 132401 (2018), the authors use a nearly identical measurement setup to ours and clearly demonstrate that the Hall voltage follows the antiferromagnetic order parameter (Fig. 3 in APL **112**(13), 132401 (2018)). We have previously replicated this data for our undoped Cr₂O₃/Pt Hall bar devices with our own setup as shown below:

Fig. R1: Temperature dependence of the Hall resistance at zero applied field after field cooling in -3T (black squares), $+3\text{T}$ (red circles), $+0.6\text{T}$ (green triangles), and -0.6T (blue triangles).

Fig. R1 shows the temperature dependence of the transverse Hall resistance ($\propto V_{xy}$) at zero magnetic field after field cooling in -3T (black squares), $+3\text{T}$ (red circles), $+0.6\text{T}$ (green triangles), and -0.6T (blue triangles). All curves merge at around 307 K , the Néel temperature of undoped Cr_2O_3 . The temperature dependence of the Hall resistance is clearly correlated with the temperature dependence of the antiferromagnetic order parameter. In fact, the temperature dependence of the Hall resistance has even been used in PRL **115**, 097201 (2015) to determine the critical exponent β . In addition to the temperature dependence, we also investigated the magnetic field dependence of the transverse and the longitudinal Hall resistance. Such investigations have been routinely performed in the literature as well. The transverse Hall resistance for fields applied along the c -axis of undoped Cr_2O_3 is known to show a weak linear field dependence for temperatures below T_N where the boundary magnetization is aligned along the c -axis and virtually saturated (see for example Fig. 2b in Phys. Rev. Appl. **13**, 034052 (2020)).

We strongly agree with the reviewer that “more evidence that the voltages induced-Hall resistance variations are distinct under different magnetic fields” is highly desirable. In order to show unambiguously that the different states of virtually zero ($V_{xy}^{zero} \approx 0$) and non-zero V_{xy} are magnetically distinct states of B-doped Cr_2O_3 we measured the magnetic field dependence of V_{xy}^{zero} and $V_{xy}^{non-zero}$.

Fig. R2: Magnetic field dependence of the transverse Hall voltage of the two states V_{xy}^{zero} and $V_{xy}^{non-zero}$ measured in a B:Cr₂O₃/Pt Hall bar device. The two distinct states are prepared by voltage pulses of +24 V (red circles) and -25 V (black squares).

Fig. R2 shows the magnetic response of V_{xy}^{zero} (red circles) and $V_{xy}^{non-zero}$ (black squares) at $T=300$ K measured in a B:Cr₂O₃/Pt Hall bar device. The two different states are initialized by voltage pulses of +24 V (selecting V_{xy}^{zero}) and -25 V (selecting $V_{xy}^{non-zero}$). V_{xy}^{zero} vs. H clearly shows a positive slope while $V_{xy}^{non-zero}$ vs. H is qualitatively distinct through the virtual absence of magnetic field dependence. This finding strongly supports the magnetically distinct behavior of the two states. The magnetic field dependence of the transverse Hall signal is consistent with spin Hall magnetoresistance. The V_{xy}^{zero} state is associated with in-plane orientation of the Néel vector and thus in-plane orientation of the boundary magnetization. As a result, the applied magnetic field normal to the surface creates maximum torque on the boundary magnetization tilting it out of the plane with increasing applied magnetic field. The increase of the normal component of the boundary magnetization increases the spin Hall magnetoresistance. The $V_{xy}^{non-zero}$ state is characterized by a large spin Hall magnetoresistance already at $H=0$ consistent with an out-of-plane orientation of the boundary magnetization. In this state, the applied magnetic field and the boundary magnetization are collinear, giving rise to minimal torque on the boundary magnetization, resulting in virtually negligible magnetic field response.

In order to include these supportive findings in the manuscript we add on page 8 at the end of the top paragraph:

Although the magnetic field has no effect on switching, it does affect the Hall voltage. The dissimilar field dependence of V_{xy} in the two AFM states provides strong support for the magnetic origin of the switching shown in Figs. 2 and 3 (for details see Fig. S5 in supplementary note 3).

We have added to the supplementary information:

Fig. S5: Magnetic field dependence of the transverse Hall voltage of the two states V_{xy}^{zero} and $V_{xy}^{non-zero}$ measured in a B:Cr₂O₃/Pt Hall bar device. The two distinct states are prepared by voltage pulses of +24 V (red circles) and -25 V (black squares).

In order to show unambiguously that the different states of virtually zero ($V_{xy}^{zero} \approx 0$) and non-zero V_{xy} are magnetically distinct states of B-doped Cr₂O₃ we measured the magnetic field dependence of V_{xy}^{zero} and $V_{xy}^{non-zero}$. Fig. S5 shows the magnetic response of V_{xy}^{zero} (red circles) and $V_{xy}^{non-zero}$ (black squares) at $T=300$ K measured in a B:Cr₂O₃/Pt Hall bar device. The two different states are initialized by voltage pulses of +24 V (selecting V_{xy}^{zero}) and -25 V (selecting $V_{xy}^{non-zero}$). V_{xy}^{zero} vs. H clearly shows a positive slope while $V_{xy}^{non-zero}$ vs. H is qualitatively distinct through the virtual absence of magnetic field dependence. This finding strongly supports the magnetically distinct behavior of the two states. The magnetic field dependence of the transverse Hall signal is consistent with spin Hall magnetoresistance. The V_{xy}^{zero} state is associated with in-plane orientation of the Néel vector and thus in-plane orientation of the boundary magnetization. As a result, the applied magnetic field normal to the surface creates maximum torque on the boundary magnetization tilting it out of the plane with increasing applied magnetic field. The increase of the normal component of the boundary magnetization increases the spin Hall magnetoresistance. The $V_{xy}^{non-zero}$ state is characterized by a large spin Hall magnetoresistance already at $H=0$ consistent with an out-of-plane orientation of the boundary magnetization. In this state, the applied magnetic field and the boundary magnetization

are collinear, giving rise to minimal torque on the boundary magnetization, resulting in virtually negligible magnetic field response.

2. The authors claim that “the switching of the antiferromagnetic order parameter is reflected in the transverse Hall voltage”, so they ascribe the variation of V_{xy} observed in their experiments to the switching of the antiferromagnetic order. However, the conclusion is not supported by the experiments. The variation of the V_{xy} can also be caused by other artifacts. Therefore, as suggested by the reviewer #1, some direct evidence may be imperative to support the conclusion.

Reply:

In addition to the evidence above, we strongly agree with the reviewer that independent confirmation for voltage-controlled reorientation of the Néel vector and the associated boundary magnetization is desirable. Therefore, we made an effort to directly probe the reorientation of the boundary magnetization via magnetic force microscopy (MFM).

Fig. R3: Top row shows the topography of a part of one uncontacted leg of the Pt Hall bar. The bottom row shows MFM phase contrasts for the pristine state (left), after application of 2 s voltage pulses of +10 V (middle) and -10 V (right).

The top row of Fig. R3 shows the topography of a part of one uncontacted leg of the Pt Hall bar measured via atomic force microscopy. The Pt Hall bar has been deposited on a B:Cr₂O₃ thin film. The bottom row shows MFM phase contrasts for the pristine state prior to application of a gate voltage (left), after application of a 2 s voltage pulse of +10 V (middle) and subsequent application of a 2 s voltage pulse of -10 V (right). Clearly visible is the drastic reduction of the MFM contrast after application of +10 V and its partial recovery after application of -10 V. We

realize that magnetic signals of antiferromagnets are small. To rule out that the contrasts originate from electrostatic long-range forces we performed additional Kelvin probe microscopy shown in Fig.R4. The top row shows the topography of the entire uncontacted Hall bar (left) together with the KPFM. The KPFM images are captured 20 minutes after application of +10 V (middle) and -10 V (right) for 2 s, respectively. The bottom figure shows the corresponding line scans of the KPFM signals as indicated by the red lines in the top KPFM images. The miniscule differences in the potential line scans between positive and negative applied voltages indicate that the contrasts displayed in Fig. 4 are magnetic in origin.

Fig. R4: The top row shows the topography of the entire uncontacted Hall bar (left) together with the KPFM. KPFM images are captured 20 minutes after application of +10 V (middle) and -10 V (right) for 2 s, respectively. The bottom figure shows the corresponding line scans of the KPFM signals as indicated by the red lines in the top KPFM images.

We esteem the independent evidence from magnetic force microscopy important enough to include Fig. 4 in the manuscript while Fig. R4 was added to the supplementary information. In the manuscript we modify the abstract by changing the sentence “Magnetometry, spin resolved inverse photoemission, electric transport measurements and piezo force microscopy reveal B-dependent T_N and resistivity enhancement, spin-canting, anisotropy reduction and dynamic polarization hysteresis.” into **Combined magnetometry, spin resolved inverse photoemission, electric transport and scanning probe microscopy measurements reveal B-dependent T_N and resistivity enhancement, spin-canting, anisotropy reduction, dynamic polarization hysteresis and gate voltage dependent orientation of boundary magnetization.**

On page 8 we include a paragraph entitled

Independent evidence for electrically controlled magnetism in B:Cr₂O₃

To provide independent evidence that the switching effects shown in Fig. 2 and Fig. 3 are magnetic in origin, we carried out additional measurements by magnetic force microscopy (MFM). MFM utilizes the long-range forces, originating from interaction between a magnetized tip and the magnetic stray field of the sample. Figure 4 shows the topographic and MFM images of a segment of the Pt Hall cross, which was deposited on the same B:Cr₂O₃ thin film used for the devices in Fig. 2 and Fig. 3. The MFM images illustrate a change in the magnetic response signal of the pristine sample after application of the poling pulses of 2 s duration and +/-10 V magnitude. Clearly visible is the drastic reduction of the MFM contrast after application of the +10 V pulse and its partial recovery after application of the -10 V pulse. Kelvin probe force microscopy measurements (Fig. S6 in the supplementary information) allowed us to rule out the effect of the parasitic electrostatic tip-sample interaction signal on the observed MFM contrast variations.

Fig. 4: Top row: topographic images of a segment of the Pt Hall cross before and after application of the poling pulses (2 s; +/-10 V). Bottom row: MFM images of the same segment after application of the poling pulses.

The supplementary information is extended by

Supplementary Note 4: Kelvin Probe Force Microscopy (KPFM)

To rule out that the MFM contrast variations in Fig. 4 originate from the electrostatic long-range forces, we performed additional Kelvin probe microscopy measurements. Figure S6 shows the topographic image of the entire Hall cross-bar along with its KPFM images acquired 20 minutes after application of +10 V and -10 V voltage pulses of 2 s duration. The miniscule difference in the cross-sectional profiles taken from these KPFM images strongly suggests no contribution of the electrostatic signal to the MFM contrast variations displayed in Fig. 4 in the main text, which, therefore, could be reliably attributed to the magnetization effect.

Fig. S6: Topographic (left) and KPFM (middle and right) images of the entire Hall cross-bar structure. The plot at the bottom shows the cross-sectional profiles of two KPFM signals taken along the horizontal red lines.

3. The authors mentioned the CMOS-compatible temperatures in the title, i.e. 300K to 400K. However, most of the experiments are conducted at room temperature. Thus maybe the title can be changed from “...at CMOS-compatible temperatures” to “... at room temperature”.

Reply: We agree with the reviewer that CMOS-compatibility is technologically very significant. The reason to investigate B-doped Cr_2O_3 goes back to our early attempts to increase the Néel temperature for device applications (Appl. Phys. Lett. **104**, 222402 (2014)). We therefore think that it is important to inform the community that B-doped Cr_2O_3 indeed can serve as a single phase material for pure voltage controlled spintronics functioning up to 400K. We show switching hysteresis in the supplementary material at 330 and 400 K. It is true that these hysteresis curves are not quite as nice as those shown in Fig. 3 of the main text, but hysteretic switching has clearly been demonstrated up to 400 K.

We would like to keep the CMOS compatibility in the abstract and throughout the manuscript but agree to shorten the title of the manuscript which now reads:

Voltage controlled Néel vector rotation in zero magnetic field

4. The author ascribe the “ $V_{xy} \approx 0$ and $V_{yx} \approx -15\text{mV}$ ” to 90-degree rotation of the Néel vector, why?

Reply: A qualitative argument is provided by a comparison between the switching of the transverse Hall resistance in a system where the Néel vector is known to reverse between two degenerate 180 degree single domain states (undoped Cr_2O_3) and the system under investigation

Fig. R5: Comparison between the symmetric switching of the transverse Hall signal in an undoped $\text{Cr}_2\text{O}_3/\text{Pt}$ Hall bar device (right panel), where the applied electric field needs to be accompanied by an applied magnetic field, and the asymmetric switching of the transverse Hall signal in a B: $\text{Cr}_2\text{O}_3/\text{Pt}$ Hall bar device (left panel).

here, *i.e.*, B: Cr_2O_3 . The comparison is illustrated in the subsequent figure Fig. R5.

The right panel of Fig. R5 shows the switching of the Hall resistance on application of voltage pulses and simultaneous presence of an applied magnetic field of 1 T in an undoped $\text{Cr}_2\text{O}_3/\text{Pt}$ Hall bar device. There is no dispute about the well-established fact that the Néel vector in undoped Cr_2O_3 switches by 180 degrees. As a result of the 180 degree Néel vector rotation the boundary magnetization switches by 180 degree as well. In accordance with spin Hall magnetoresistance but also in accordance with all potential Hall resistance mechanisms, which are odd in magnetization, the sign of V_{xy} changes on reversal of the magnetization as shown in the lower part of the right panel of Fig. R5. This particular symmetry of the Hall resistance signal completely changes in the case of B: $\text{Cr}_2\text{O}_3/\text{Pt}$ Hall bar devices. The left panel of Fig. R5 shows again the switching behavior observed in our B: $\text{Cr}_2\text{O}_3/\text{Pt}$ Hall bar devices. The most prominent feature is the asymmetry in the switching between a state of virtually zero transverse Hall voltage and a state of non-zero transverse Hall voltage. Clearly this behavior is not expected as a result of 180 degree reversal of the boundary magnetization. The zero state strongly suggests in-plane rather than out of plane orientation of the boundary magnetization. The added experimental evidence from the magnetic field dependence of the transverse Hall signal supports this interpretation. As outlined above, a sizable magnetic response is expected when the applied magnetic field is oriented perpendicular to the boundary magnetization. It is also worth to mention that it is the state of virtual zero transverse Hall signal which shows the strong dependence on an applied magnetic field. This fact rules out other non-magnetic mechanisms, which may give rise to a false signal.

5. Spin Hall resistance in the B-doped Cr₂O₃/Pt system should be studied before the voltage controlled experiments.

Reply: We agree with the reviewer that more detailed knowledge of the magnetic field dependence of the Hall resistance in B:Cr₂O₃/Pt Hall bar devices is highly desirable. Although there is certainly room for many additional investigations, we have added several important missing pieces of experimental evidence and performed magnetic field dependent measurements of the transverse Hall signal in B:Cr₂O₃/Pt Hall bar devices. The results detailed in Fig. S5 of the supplementary note 3 make the case for switching between two orthogonal magnetic states much stronger. The same holds for the additional evidence from MFM investigations. It is worth to mention that the field dependence of the transverse Hall signal in the non-zero state is consistent with the field dependence of the transverse Hall signal in undoped Cr₂O₃/Pt Hall bar devices significantly below the Néel temperature. We interpret this similarity by collinear alignment of the boundary magnetization with respect to the applied magnetic field for Cr₂O₃/Pt Hall bar devices in general and B:Cr₂O₃/Pt Hall bar devices in the non-zero state.

6. To better understand the mechanism of the variation of V_{xy} , the authors should present more control experiences. For instance, when the oxide is not B-doped Cr₂O₃, but other AFM or even non-magnetic materials, whether the resistance variation still exists.

Reply: As mentioned in our reply to point number 2 the mechanism of variation of V_{xy} has been extensively studied in the literature and by us. The most relevant control experiment is the bulk of literature available on undoped Cr₂O₃/heavy metal (Pt, Ta, W) Hall bar structures. Please also have a look at our control experiments performed on undoped Cr₂O₃/Pt shown in Fig.R1 of this reply and the supplementary information where we show the piezo force microscopy results on undoped Cr₂O₃. Isothermal switching of the Hall resistance has been demonstrated in Kosub, T. *et al.* Purely antiferromagnetic magnetoelectric random access memory. *Nature Communications* **8**, 13985, doi:10.1038/ncomms13985 (2017). However, this switching was based on the magnetoelectric properties of chromia and therefore requiring an applied magnetic field of $H=0.5\text{MA/m}$ (0.63T). It is the major point of our manuscript that switching of the Hall resistance is achieved in the absence of an applied magnetic field. The switching mechanism is therefore different (not related to the linear magnetoelectric response) and the implications for device applications are very significant. Pure voltage-control of antiferromagnetic states is one of the major goals of antiferromagnetic spintronics. It is of major significance, but of course there is not much success reported in the literature. Among the more recent successes are Néel spin orbit torque experiments which require current. Switching in the absence of current is what the community is still striving for to achieve ultra-low power spintronic applications.

7. Besides, the voltage controlled Néel vector rotation has been realized in many previous works.

Reply: We agree with the reviewer that there are many reports and claims of Néel vector rotation. Those include Néel vector rotation by Néel spin-orbit torque mechanism (*Science* **351** (6273), 587-590 (2016), *Nature Communications* **9** (1), 348 (2018).), Néel vector rotation in single phase multiferroics either require a magnetic field or is limited at best to a few cycles.

Néel vector rotation in two-phase multiferroics, which are composites of typically piezoelectric and magnetoelastic antiferromagnets, is plagued by fabrication issues, scaling, material fatigue and often lacks nonvolatility (Spaldin, N. A. & Ramesh, R. Advances in magnetoelectric multiferroics. *Nat Mater* **18**, 203-212, doi:10.1038/s41563-018-0275-2 (2019)). To the best of our knowledge, the most desirable Néel vector rotation, which is voltage rather than current controlled, repeatable, requires no magnetic field, is not detected via exchange bias, and happens in a single-phase material at temperature above 300 K has never been shown prior to our work.

Reviewer #4 (Remarks to the Author):

Authors report the voltage-driven switching of the magnetoresistive resistance in Pt on B-doped Cr₂O₃. They attributed to the results to the piezoelectric switching of the polar nanoregion (PNR) and the accompanied rotation of boundary magnetization. I recognize the effort to revise the manuscript from the previous version. But still, there are some important concern on the proposed mechanism. Although various properties of the B-doped Cr₂O₃, some of them are heart of this paper, are known on the analogy of the un-doped Cr₂O₃, it is not obvious. Based on them, I think the paper is not acceptable in the present form.

Reply: We are thankful that the reviewer recognizes our effort to revise the manuscript from the previous version. We agree that there are still weaknesses in the manuscript he has inspected. To resolve the remaining concerns we are happy to present now independent data from magnetic force microscopy. Together with our additional transport investigations on the magnetic field dependence of the transverse Hall voltage in the two distinct magnetic states we are convinced that we made a strong case for our proposed switching mechanism.

As pointed out by reviewer 1, authors attribute the resistive switching to the rotation of the boundary magnetization. Authors claim that the coupling between the boundary magnetization and the Hall effect should be robust for the B-doped Cr₂O₃, but it seems that this claiming is just the speculation from the analogy of the un-doped Cr₂O₃.

Reply: It is true that we extrapolate the interpretation of the Hall resistance data of B-doped chromia from the observations of undoped chromia. However, we do have direct evidence that B-doped chromia has boundary magnetization just as undoped chromia which is directly coupled to the magnetoelectrically controllable antiferromagnetic order parameter. This evidence is given in Fig. 1 c of the manuscript and in our previous work Street, M. et al. Increasing the Néel temperature of magnetoelectric chromia for voltage-controlled spintronics. *Appl Phys Lett* **104**, doi:10.1063/1.4880938 (2014). In addition it is worth to mention that boundary magnetization is a property of those antiferromagnetic materials which fulfill the symmetry requirements of a linear magnetoelectric effect. It's presence does not hinge on the strength of the linear magnetoelectric susceptibility (Belashchenko, K. D. Equilibrium Magnetization at the Boundary of a Magnetoelectric Antiferromagnet. *Phys Rev Lett* **105**, 147204 (2010).) Most important, however, our MFM data and the additional transport data strongly reinforce our claims.

In order to show unambiguously that the different states of virtually zero ($V_{xy}^{zero} \approx 0$) and non-zero V_{xy} are magnetically distinct states of B-doped Cr_2O_3 we measured the magnetic field dependence of V_{xy}^{zero} and $V_{xy}^{non-zero}$.

Fig. R2: Magnetic field dependence of the transverse Hall voltage of the two states V_{xy}^{zero} and $V_{xy}^{non-zero}$ measured in a B: Cr_2O_3 /Pt Hall bar device. The two distinct states are prepared by voltage pulses of +24 V (red circles) and -25 V (black squares).

Fig. R2 shows the magnetic response of V_{xy}^{zero} (red circles) and $V_{xy}^{non-zero}$ (black squares) at $T=300$ K measured in a B: Cr_2O_3 /Pt Hall bar device. The two different states are initialized by voltage pulses of +24 V (selecting V_{xy}^{zero}) and -25 V (selecting $V_{xy}^{non-zero}$). V_{xy}^{zero} vs. H clearly shows a positive slope while $V_{xy}^{non-zero}$ vs. H is qualitatively distinct through the virtual absence of magnetic field dependence. This finding strongly supports the magnetically distinct behavior of the two states. The magnetic field dependence of the transverse Hall signal is consistent with spin Hall magnetoresistance. The V_{xy}^{zero} state is associated with in-plane orientation of the Néel vector and thus in-plane orientation of the boundary magnetization. As a result, the applied magnetic field normal to the surface creates maximum torque on the boundary magnetization

tilting it out of the plane with increasing applied magnetic field. The increase of the normal component of the boundary magnetization increases the spin Hall magnetoresistance. The $V_{xy}^{non-zero}$ state is characterized by a large spin Hall magnetoresistance already at $H=0$ consistent with an out-of-plane orientation of the boundary magnetization. In this state, the applied magnetic field and the boundary magnetization are collinear, giving rise to minimal torque on the boundary magnetization, resulting in virtually negligible magnetic field response.

In addition to the evidence above, we strongly agree with the reviewer that independent confirmation for voltage-controlled reorientation of the Néel vector and the associated boundary magnetization is desirable. Therefore, we made an effort to directly probe the reorientation of the boundary magnetization via magnetic force microscopy (MFM).

Fig. R3: Top row shows the topography of a part of one uncontacted leg of the Pt Hall bar. The bottom row shows MFM phase contrasts for the pristine state (left), after application of 2 s voltage pulses of +10 V (middle) and -10 V (right).

The top row of Fig. R3 shows the topography of a part of one uncontacted leg of the Pt Hall bar measured via atomic force microscopy. The Pt Hall bar has been deposited on a B:Cr₂O₃ thin film. The bottom row shows MFM phase contrasts for the pristine state prior to application of a gate voltage (left), after application of a 2 s voltage pulse of +10 V (middle) and subsequent application of a 2 s voltage pulse of -10 V (right). Clearly visible is the drastic reduction of the MFM contrast after application of +10 V and its partial recovery after application of -10 V. We realize that magnetic signals of antiferromagnets are small. To rule out that the contrasts originate from electrostatic long-range forces we performed additional Kelvin probe microscopy shown in Fig.R4. The top row shows the topography of the entire uncontacted Hall bar (left) together with the KPFM. The KPFM images are captured 20 minutes after application of +10 V (middle) and -10 V (right) for 2 s, respectively. The bottom figure shows the corresponding line scans of the KPFM signals as indicated by the red lines in the top KPFM images. The miniscule differences in the potential line scans between positive and negative applied voltages indicate that the contrasts displayed in Fig. 4 are magnetic in origin.

Fig. R4: The top row shows the topography of the entire uncontacted Hall bar (left) together with the KPFM. KPFM images are captured 20 minutes after application of +10 V (middle) and -10 V (right) for 2 s, respectively. The bottom figure shows the corresponding line scans of the KPFM signals as indicated by the red lines in the top KPFM images.

We implemented substantial changes in the manuscript and the supplementary information based on the above. Those changes are given in detail in the reply to reviewer 1 and are also highlighted in the revised manuscript and supplementary information in green.

Besides, the mechanism of the hall effect of Pt on Cr₂O₃ is under the debate. For example, other than the boundary magnetization, there are some reports proposing other mechanism such as Spin Hall effect (R. Schulitz et al., Appl. Phys. Lett. 112, 132401 (2018).), and interfacial spin chirality (T. Moriyama et al., Phys. Rev. Applied 13, 034052 (2020).)

Reply: We agree with the reviewer that the Hall effect in Cr₂O₃/PT systems is still under debate. We state that in the manuscript. Although we have a propensity for the spin Hall magnetoresistance mechanism, it has been established that whatever the mechanism for V_{xy} is, V_{xy} is a reliable proxy for the boundary magnetization (see for example our own data provided in this reply in Fig. R1). We like to point out that the boundary magnetization is not at all a competing mechanism to spin Hall magnetoresistance. On the contrary, it is the boundary magnetization which, via the mechanism of the inverse spin Hall effect, gives rise to V_{xy} in the

model of spin Hall magnetoresistance. The spin chirality falls into the category of alternative interpretation together with the referenced anomalous Hall effect from proximity. As pointed out above and in the original manuscript the details are controversial. This manuscript will not resolve this controversy. However, the fact remains experimentally established that the orientation of the Néel vector can be inferred from the transport data. It is important to stress that the details of the mechanism do not matter for our purposes as long as the transverse Hall signal is correlated with the orientation of the boundary magnetization. Our addition of magnetic field dependence of V_{xy} in the two distinct magnetic states excludes a non-magnetic interpretation of the data shown in Fig. 2 and 3 of the manuscript. The same is true for the addition of the MFM data.

In order to acknowledge that in addition to spin Hall magnetoresistance and proximity effect there are other possible mechanism for the Hall resistance we replace on page 5 the sentence:

“A potential additional contribution originates from the anomalous Hall effect caused by magnetization which is proximity induced in the heavy metal Hall bar by the exchange field of the boundary magnetization.” by

Potential additional contributions might originate from the anomalous Hall effect caused by magnetization which is proximity induced in the heavy metal Hall bar by the exchange field of the boundary magnetization or by anomalous Hall effect generated by spin chirality (T. Moriyama et al., Phys. Rev. Applied 13, 034052 (2020)).

To ensure the proposed mechanism, the direct evidence to show the switching of the Néel vector and the boundary magnetization by the electric field.

Reply: We agree with the reviewer and are convinced that our MFM and additional transport data make a compelling case.

In addition, as discussed in the un-doped Cr₂O₃, the boundary magnetization of the partial effect of the magnetoelectricity of Cr₂O₃. However, I cannot find the robust evidence that the B-doped Cr₂O₃ also show the similar magnetoelectricity to the un-doped Cr₂O₃.

Reply: We understand that reviewer missed the evidence that B-doped chromia is magnetoelectric in a similar way undoped chromia is. We did provide the evidence in Fig. 1 c and the corresponding interpretation of the spin polarized inverse photoemission data. It is worth noting that spin polarized inverse photoemission is extremely surface sensitive and that the observed spin splitting after magnetoelectric annealing provides strong evidence for the reversal of the boundary magnetization intimately tied to the antiferromagnetic order parameter (Belashchenko, K. D. Equilibrium Magnetization at the Boundary of a Magnetoelectric

Antiferromagnet. *Phys Rev Lett* **105**, 147204 (2010)). The boundary polarization at the surface, as measured by spin polarized inverse photoemission, reverses sign when the E.B product is reversed as has been previously demonstrated [M. Street, Will Echtenkamp, Takashi Komesu, Shi Cao, P. A. Dowben, and Ch. Binek, “Increasing the Néel Temperature of Magnetoelectric Chromia for Voltage-Controlled Spintronics”, *Applied Physics Letters* **104** (2014) 222402; doi: 10.1063/1.4880938]. Thus the case for a magneto-electric B-doped chromia is compelling. The reason that we did not stress this point very much in the manuscript originates from the fact that the linear magnetoelectric effect, although present, cannot be responsible for the reorientation of the boundary magnetization in the case of zero applied magnetic field. The mechanism is different as outlined in the manuscript. It is this difference, which makes our finding exciting.

As pointed out in the manuscript, many of physical/structural properties such as magnetic anisotropy, spin orientation, local circumstance of Cr³⁺ are different from the un-doped Cr₂O₃. Then, the similar magnetoelectricity should not be ensured in the B-doped Cr₂O₃. For example, for the B-doping, the valence state of Cr ion can be change to keep the electrical neutrality. Thus, authors have to show that the B-doped C₂O₃ actually show the magnetoelectricity similar to the un-doped Cr₂O₃ and the existence of the boundary magnetization on the B-doped Cr₂O₃(0001).

Reply: As addressed above we presented evidence for the magnetoelectricity of B-doped chromia. However, the linear magnetoelectric effect as known from pure chromia cannot be relevant for the observed voltage-controlled magnetization reversal in zero applied magnetic field. It is true that the effects of B-doping are rather complex also due to the fact that B can be found in multiple charge states. As a matter of fact there is a very detailed electron energy loss spectroscopy analysis performed on B:Cr₂O₃ films grown by us. The spectroscopy data have been paired with density functional theory, which finds the presence of functional BCr₄ tetrahedra (C. Sun et al., *Microsc. Microanal.* 23 (Suppl 1), 2017) and BO₃ triangles confirming our model for polar nanoregions (see Fig. 5 in the manuscript.)

In order to make this point clear we modify the manuscript. On page 9 we replace the sentence “The local strain moves the B atom to an off-center position resulting in emergence of PNRs.” by:

The local strain moves the B atom to an off-center position within BCr₄ tetrahedra (C. Sun et al., *Microsc. Microanal.* 23 (Suppl 1), 2017) resulting in emergence of PNRs⁵⁰.

Reviewers' Comments:

Reviewer #1:

Remarks to the Author:

The manuscript describes a voltage-induced switching of Hall resistance in B-doped Cr₂O₃/Pt bilayers at room temperature. The quality of the work is somehow improved with the revision. But my main concern is that why the Hall resistance changes are related to the Néel vector switching of antiferromagnetic moments. Thus I recommend the authors to address the following questions and comments before the consideration of publication.

(1) It is still unclear that the Hall resistance changes are related to the Néel vector switching of antiferromagnetic moments or related to the boundary magnetization? What is the relationship between boundary magnetization and Néel vector? How thick of the magnetic ordering is switched during the switching?

(2) If the voltage induced Néel vector rotation, the switching direction should be clarified. More evidence on the Néel vector rotation direction (such as XMLD) should be helpful.

(3) Whether the voltage control experiments can be generalized to some other antiferromagnetic systems? Which parameter is necessary for the present observation?

Reviewer #3:

Remarks to the Author:

I recognize the authors' effort to justify their claims. The additional data (MFM, KPFM etc) strengthen the experimental support to their claim. Now, I recommend the publication.

Letter Legend: Referee comments are given in **Blue**, our responses in **Black** and the corresponding changes to the manuscript are in **Green**.

Reviewers' comments:

Reviewer #1 (Remarks to the Author):

The manuscript describes a voltage-induced switching of Hall resistance in B-doped Cr₂O₃/Pt bilayers at room temperature. The quality of the work is somehow improved with the revision. But my main concern is that why the Hall resistance changes are related to the Néel vector switching of antiferromagnetic moments. Thus I recommend the authors to address the following questions and comments before the consideration of publication.

(1) It is still unclear that the Hall resistance changes are related to the Néel vector switching of antiferromagnetic moments or related to the boundary magnetization? What is the relationship between boundary magnetization and Néel vector? How thick of the magnetic ordering is switched during the switching?

Reply: We are happy to read that also reviewer #1 thinks that the quality of the revised manuscript improved and that the reviewer now sees a pathway towards publication of the manuscript. We agree that the Hall measurements as well as the MFM measurements detect reorientation of the boundary magnetization and not directly reorientation of the Néel vector. However, the intimate relation between boundary magnetization and orientation of the Néel vector is in general established for magnetoelectric antiferromagnets. In the manuscript we briefly refer to the bulk of work that has been done in this context by stating “However, in ME antiferromagnets, an equilibrium magnetic moment associated with the antiferromagnetic order parameter is symmetry allowed.¹⁴⁻¹⁶ This moment can be sizable even in the presence of roughness and enables effective coupling with the magnetization of an adjacent ferromagnet.” We agree that we should elaborate on this important fact in the revised manuscript. It is well known from rigorous symmetry considerations that interfaces and surfaces of linear magnetoelectric antiferromagnets have a symmetry allowed boundary magnetization. Its orientation is directly coupled with the orientation of the Néel vector (Belashchenko, K. D. *Equilibrium Magnetization at the Boundary of a Magnetoelectric Antiferromagnet*. Phys Rev Lett **105**, 147204 (2010); Andreev, A. F. *Macroscopic magnetic fields of antiferromagnets*. Journal of Experimental and Theoretical Physics Letters **63**, 758-762, doi:10.1134/1.566978 (1996).). The theoretically predicted relation between boundary magnetization and orientation of the antiferromagnetic order parameter has been experimentally confirmed for instance using XMCD to image the boundary magnetization associated with a particular antiferromagnetic domain orientation (Wu, N. et al. *Imaging and Control of Surface Magnetization Domains in a Magnetoelectric Antiferromagnet*. Phys Rev Lett **106**, doi:https://doi.org/10.1103/PhysRevLett.106.087202 (2011).). In our manuscript we have shown

via spin resolved inverse photoemission (see inset c in Fig.1) that B-doped chromia is a linear magnetoelectric. Previously it had been shown that the antiferromagnetic domain state can be selected by magnetoelectric annealing selecting in turn the orientation of the boundary magnetization (Street, M. et al. *Increasing the Néel temperature of magnetoelectric chromia for voltage-controlled spintronics*. Appl Phys Lett **104**, doi:10.1063/1.4880938 (2014).). Therefore evidence for the reorientation of boundary magnetization is evidence for the reorientation of the Néel vector.

Considering the chain of cause and effect one has to look at the switching of the boundary magnetization in the following way. First, the applied field reorients the Néel vector. Because the applied field is homogeneous, the reorientation of the Néel vector takes place homogeneously throughout the entire sample. The boundary magnetization is strictly tied to the orientation of the Néel vector and follows its reorientation.

In order to stress the intimate connection between boundary magnetization and Néel vector orientation in the manuscript, we add on page 2 after “However, in ME antiferromagnets, an equilibrium magnetic moment associated with the antiferromagnetic order parameter is symmetry allowed.¹⁴⁻¹⁶”: *Note that due to the rigorous symmetry argument leading to boundary magnetization in linear magnetoelectric antiferromagnets, the boundary magnetization is strictly tied to the orientation of the Néel vector.*

(2)If the voltage induced Néel vector rotation, the switching direction should be clarified. More evidence on the Néel vector rotation direction (such as XMLD) should be helpful.

Reply: We agree with the reviewer that the orientation of the Néel vector and boundary magnetization is an important question. Because the strict connection between boundary magnetization and Néel vector is well established for magnetoelectric antiferromagnets (see references 14 and 16 for theory and Ref. 15 for XMCD measurements) we made a major effort to evidence the reorientation of the boundary magnetization. We go into details in the supplementary note 3. Of particular importance for the reply to the question here is the paragraph in the supplementary note 3 highlighted here in red.

Fig. S5: Magnetic field dependence of the transverse Hall voltage of the two states V_{xy}^{zero} and $V_{xy}^{non-zero}$ measured in a B:Cr₂O₃/Pt Hall bar device. The two distinct states are prepared by voltage pulses of +24 V (red circles) and -25 V (black squares).

In order to show unambiguously that the different states of virtually zero ($V_{xy}^{zero} \approx 0$) and non-zero V_{xy} are magnetically distinct states of B-doped Cr₂O₃ we measured the magnetic field dependence of V_{xy}^{zero} and $V_{xy}^{non-zero}$. Fig. S5 shows the magnetic response of V_{xy}^{zero} (red circles) and $V_{xy}^{non-zero}$ (black squares) at $T=300$ K measured in a B:Cr₂O₃/Pt Hall bar device. The two different states are initialized by voltage pulses of +24 V (selecting V_{xy}^{zero}) and -25 V (selecting $V_{xy}^{non-zero}$). V_{xy}^{zero} vs. H clearly shows a positive slope while $V_{xy}^{non-zero}$ vs. H is qualitatively distinct through the virtual absence of magnetic field dependence. This finding strongly supports the magnetically distinct behavior of the two states. The magnetic field dependence of the transverse Hall signal is consistent with spin Hall magnetoresistance.

The V_{xy}^{zero} state is associated with in-plane orientation of the Néel vector and thus in-plane orientation of the boundary magnetization. As a result, the applied magnetic field normal to the surface creates maximum torque on the boundary magnetization tilting it out of the plane with increasing applied magnetic field. The increase of the normal component of the boundary magnetization increases the spin Hall magnetoresistance. The $V_{xy}^{non-zero}$ state is characterized by a large spin Hall magnetoresistance already at $H=0$ consistent with an out-of-plane orientation of the boundary magnetization. In this state, the applied magnetic field and the boundary magnetization are collinear, giving rise to minimal torque on the boundary magnetization, resulting in virtually negligible magnetic field response.

To make clear that the data in Fig. S5 allows the assignment of magnetization states to the distinct voltage states we extend on page 8 “The dissimilar field dependence of V_{xy} in the two antiferromagnetic states provides strong support for the magnetic origin of the switching shown in Figs. 2 and 3 (for details see Fig. S5 in supplementary note 3).” into “The dissimilar field

dependence of V_{xy} in the two antiferromagnetic states provides strong support for the magnetic origin of the switching shown in Figs. 2 and 3 and allows the assignment of the orientation of the boundary magnetization to the distinct voltage states (for details see Fig. S5 in supplementary note 3).”

(3) Whether the voltage control experiments can be generalized to some other antiferromagnetic systems? Which parameter is necessary for the present observation?

Reply: We thank the reviewer for bringing up this interesting question. It gives us another opportunity to highlight the beneficial properties of our material when compared to the very few available potential competing materials. Perhaps the only single phase material with competing properties are strain engineered thin films of BiFeO_3 (BFO). In sharp contrast to B-doped chromia, which has only a single spontaneous order parameter, BFO has spontaneous antiferromagnetic and spontaneous ferroelectric order. The energy it takes to switch spontaneous ferroelectric order is proportional to the magnitude of the saturation polarization. Because B-doped chromia doesn't have spontaneous polarization but only electric field induced polarization, there is potential in B-doped chromia for switching at ultra-low energies. The key parameters for a system such as B-doped chromia are the following:

- Presence of antiferromagnetic order and boundary magnetization tied to the antiferromagnetic order via the symmetry conditions of broken time inversion and parity while their combined operation leaves the system invariant.
- Absence of spontaneous polarization but presence of high dielectric polarizability and, related to that, very good dielectric insulating properties to enable the presence of strong electric fields in the absence of leakage currents.
- Coupling between induced polarization and the antiferromagnetic order parameter for instance via strain.

We believe that any system satisfying these key properties is a candidate for pure voltage controlled reorientation of the Néel vector with possibility for scalable readout via Hall-like signals. That said, to the best of our knowledge, B-doped Cr_2O_3 is the first of such systems investigated.

Reviewer #3 (Remarks to the Author):

I recognize the authors' effort to justify their claims. The additional data (MFM, KPFM etc) strengthen the experimental support to their claim. Now, I recommend the publication.

Reply: We appreciate that the reviewer acknowledges our effort and are delighted to read that publication is recommended.

Reviewers' Comments:

Reviewer #1:

Remarks to the Author:

After the second iteration, the work is greatly improved. I think it is ready for the publication.